# A Novel Testing Method for Examining Corrosion Behavior of Reinforcing Steel in Simulated Concrete Pore Solutions

**DOI:** 10.3390/ma13235327

**Published:** 2020-11-24

**Authors:** Yanru Li, Jiazhao Liu, Zhijun Dong, Shaobang Xing, Yajun Lv, Dawang Li

**Affiliations:** 1Guangdong Provincial Key Laboratory of Durability for Marine Civil Engineering, Shenzhen University, Shenzhen 518060, China; liyanru6885@126.com (Y.L.); darkdanking@126.com (Y.L.); 2Shenzhen Luohu District Bureau of Construction Works, Shenzhen 518060, China; Liujz9818@126.com; 3Shenzhen Institute of Information Technology, Shenzhen 518172, China; dongzj@sziit.edu.cn; 4Hong Kong Huayi Design Consultants (S.Z) Ltd., Shenzhen 518060, China; shaobangxing@126.com

**Keywords:** reinforcing steel, corrosion, chloride, experiment, bow shaped device, concrete

## Abstract

In this paper, a new mechanical-based experimental method is proposed to determine the corrosion initiation and subsequent corrosion behavior of steel in simulated concrete pore solutions. The proposed experiment is used to investigate the corrosion of the steel wire under various different conditions and to examine the effects of pre-stress level in steel wire, passivation time of steel wire, composition and concentration of simulated concrete pore solution on the corrosion initiation, and subsequent corrosion development in the steel wire. The experimental results show that the reduction rate of the cross-section area of the steel wire increases with the increase of chloride concentration or decrease of pH value in the solution. However, for the case where the chloride concentration is high and the pH value is low, there is a slight decrease in the corrosion rate due to the coating function of the corrosion products surrounding the wire.

## 1. Introduction

Metal corrosion is a common phenomenon which can cause serious deterioration of the service performance of engineering structures. Reinforcing steel corrosion, frequently found in concrete structures exposed to chloride environments, is a typical example [1,2]. Reinforcing steel corrosion can be defined as an electrochemical process of a metal in relation to its surrounding environment, which can be represented by two electrochemical reactions of the dissolution of iron at anodic sites and the corresponding oxygen reduction at cathodic sites. The chemical reactions of corrosion are the same no matter the passive layer around steel breaks down by chloride attack or by carbonation. When depassivation occurs, areas of rust will start appearing on the affected area of the steel surface in the presence of water and oxygen. The affected steel areas become anodic whilst the passive areas are cathodic. A corrosion cell develops on steel surface because of the difference in electrical potential between the anodic and cathodic areas, while a current flows from the anode to the cathode transported by the ions in the electrolyte. The faster the solid iron is converted to ions, the greater the corrosion and the larger the current flowing in the corrosion cell. The detailed description of the electrochemical process of the corrosion of steel bars in concrete can be found in technical books, for example, [3,4,5].

Reinforcing steel embedded in concrete is usually in a passive state due to the high alkalinity of concrete pore solution, which protects the steel bars from corrosion in some extent. However, the passive state could disappear if the concrete is exposed to a marine environment since chloride ions can penetrate into concrete and cause rupture of the passive layer [6,7]. Numerous studies exist in literature on chloride-induced corrosion of reinforcement in concrete under different conditions. Angst and Vennesland [8] and Angst et al. [9] presented excellent review articles, which not only described the concept of the chloride threshold value (CTV) and discussed the factors that affect the CTV, but also assessed various measurement techniques for reinforcing corrosion. Recently, Green [10] also presented a review paper, which described the fundamental and mechanistic aspects of the protection afforded to steel reinforcement by concrete, corrosion of steel reinforcement, corrosion products composition and development, chloride- and carbonation-induced corrosion mechanisms, leaching induced corrosion of reinforcement and reinforcing steel stray current corrosion and interference. 

In the literature, two types of methods have been reported to determine the CTV. One is from a scientific point of view, in which the CTV is defined as the chloride content required for depassivation of the steel [11,12,13,14]. The other is from an engineering point of view, in which the CTV is taken as the chloride content associated with visible corrosion of steel or acceptable deterioration of a RC structure [3,4,7,11]. Note that, as far as the durability is concerned, the first definition is more related to materials which is considered to be in the initiation stage of corrosion; whereas the second one is more associated with structures which is considered to be in the propagation stage of corrosion. These two definitions could lead to different CTVs. In general, the engineering definition leads to a higher CTV and results in a large scatter of CTVs [12]. Simulated concrete pore solutions have been used to study the corrosion initiation of reinforcing steel by using laboratory testing methods [15,16]. Many researchers have examined the effect of surface finishing of steel bars on their corrosion initiation in simulated concrete pore solutions with different pH values [17,18,19,20]. Testing methods for on-site corrosion monitoring of reinforced concrete structures have been reported [21,22,23]. 

There are many electrochemical techniques that could be used to determine the CTV associated with the depassivation and subsequent corrosion behavior of reinforcing steel bars in concrete, such as monitoring the macrocell current between an anode and a cathode [24], measuring the half-cell potential [25], measuring the polarisation resistance [26], and measuring the electrochemical impedance spectroscopy [27]. However, most of these methods do not have good repeatability. Also, the CTVs obtained using different methods are found to be quite different. For example, according to Angst and Vennesland [8] the CTV varies from 0.02% to 3.08% total chloride by binder weight. Recently, Meira et al. [28] reported that average CTV ranges from 1.82% to 2.45% of cement weight for laboratory measurements and between 0.88% and 1.58% of cement weight for field exposure experiments. They concluded that the large difference could be attributed to the environmental interaction.

The wide spread of results on CTV reported in literature may be attributed to the different definitions for the corrosion initiation, the techniques used to determine the CTV [8], the surface finishing of the steel bars [17], the concrete properties and the aggressiveness of the environment [29]. In this paper a new experimental method of using a novel bow shaped device is proposed. The experimental device is then used to investigate the corrosion of the steel wire under various different conditions. The effects of pre-stress level in steel wire, passivation time of steel wire, composition and concentration of simulated concrete pore solution on the corrosion initiation and subsequent corrosion development in the steel wire are examined and a detailed discussion is provided.

## 2. Method and Device 

A new bow shaped device is described herein, which is used for examining steel wire corrosion in a simulated concrete pore solution along with the corresponding cross-section area reduction rate of the wire induced by steel corrosion. 

Consider the corrosion problem of a steel wire of uniform cross-section, subjected to a tensile force *F* when it is immersed in a corrosive solution. Assume that the steel wire has had a passivation treatment and there is a thin passivation film surrounding its surface. Let *A*_0_ be the initial cross-section area of the steel wire and *σ*_ult_ be the ultimate tensile strength of the steel wire. 

In general, the corrosion process of the steel wire after it is immersed in the corrosive solution can be divided into two stages. One is the corrosion initiation stage during which the passivation film is destroyed. At this stage, there is no actual corrosion taking place in the steel and thus the cross-section area of the steel wire remains unchanged. This stage ends when the passivation film is destroyed and the steel is about to start to corrode. The other is the corrosion stage during which the steel wire corrodes continuously. At this stage, the cross-section area of the steel wire decreases gradually with increasing time. This stage ends when the steel wire breaks due to the reduction of its cross-section area caused by the corrosion and the corroded steel wire is no longer able to sustain the applied force *F*. 

Let *t_s_* be the time at the end of the corrosion initiation stage and *t_r_* be the time at the end of the corrosion stage. Thus, the average reduction rate of the cross-section area of the steel wire can be expressed as follows,
(1)$$v=\frac{A_{0}-A(t_{r})}{t_{r}-t_{s}}$$
where *v* is the corrosion reduction rate of the cross-section area of the steel wire and *A*(*t_r_*) is the residual cross-section area of the corroded steel wire at the break time. Note that *A*(*t_r_*) can be expressed in terms of the applied force, that is *A*(*t_r_*) = *F/σ*_ult_ = *A*_0_*F/F*_max_, where *F*_max_ = *σ*_ult_*A*_0_ is the maximum force that the steel wire can sustain. Thus, Equation (1) can be rewritten as follows:(2)$$v=\frac{A_{0}-\frac{F}{\sigma_{\mathrm{ult}}}}{t_{r}-t_{s}}=\frac{A_{0}}{t_{r}-t_{s}}\cdot (\frac{F_{\max}-F}{F_{\max}})$$

From the above equation, the following equation can be obtained,
(3)$$t_{r}-t_{s}=\frac{A_{0}}{v}\cdot (\frac{F_{\max}-F}{F_{\max}})$$

Note that, for given environment conditions the average corrosion reduction rate of the cross-section area of the steel wire should be a finite number. Thus, it can be found from Equation (3) that when *A*_0_→0 or *F*→*F*_max_, *t_r_*→*t_s_*. This indicates that the break time of the steel wire could be used to represent the corrosion initiation time if the diameter of the steel wire were very small or the applied force were very close to the maximum force of the steel wire.

The concept described above can be fulfilled by using the experiment of a bow shaped device as shown in Figure 1a, in which the frame of the bow can be obtained by bending a straight plexiglass ruler and the wire is a steel wire that is to be tested for corrosion. The dimensions of the bow and the length of the wire can be adjusted based on the pre-tensile stress of the wire required in the test. After the plexiglass ruler is bent to a pre-calculated curvature, the steel wire is fixed to the small holes predrilled on the plexiglass ruler near its two ends. Copper conductive wires are connected to the ends of the steel wire at the fixing points. The other ends of the copper wires are connected to a data recorder or electrochemical workstation, as shown in Figure 1, to monitor the change of signals and/or record the times at which the steel wire starts corroding and/or breaks after it is immersed in the corrosive solution.

In order to make sure the corrosion of the steel wire occurs in the middle part of the wire after it is immersed in the corrosive solution, the two ends of the steel wire near the fixing points are coated with epoxy resin. The time is recorded immediately after the bow is immersed in the solution. After the steel wire is submerged in the corrosive solution, its passivation film will be destroyed first and then corrosion starts. How fast this process takes place is dependent on the corrosive solution used in the test. Nevertheless, when a visible corrosion is observed in the pre-stressed wire, the corresponding time is recorded as *t_s_*, whereas the time when the wire breaks is recorded as *t_r_*.

## 3. Experimental Programme

The present experimental method is developed based on the mechanical principle in which the break of a pre-stressed steel wire is purely due to the corrosion-induced increase of the stress, which reaches to the ultimate tensile strength of the wire. In the present experiment, the frame of the bow is made from plexiglass fragment of rectangular cross-section (3 mm × 40 mm). The steel wire was obtained from a Chinese company (Mingjunchangrong, Foshan, China), has a diameter of 0.1413 mm, and is made from high strength steel. The tensile test of the steel wire alone shows that the ultimate tensile strength of the steel wire can reach to 2931 MPa. The chemical compositions of the steel wire obtained from five randomly taken sampling points on the wire are shown in Table 1. Before the bow is immersed in the solution, the steel wire is first cleaned using absolute ethanol and then washed using deionized water. After it is dried, the steel wire is subjected to a passivation treatment by immersing it into a saturated Ca(OH)_2_ solution for a period of 1–31 days depending on the purpose of tests. All of the tests are conducted at room temperature (about 25 °C) in a controlled laboratory condition.

The pre-stress applied on the steel wire can be calculated based on the geometry of the deformed bow. Alternatively, it can be determined using a vibration method [30] from which the pre-force can be calculated as follows,
(4)$$F=4ml^{2}f^{2}$$
where *m* is the mass of the steel wire per-unit length, *l* is the length of the steel wire between the two fixed points, and *f* in Hz is the fundamental frequency of the steel wire in the bow shaped device, which can be obtained by using the laser doppler vibrometer (LDV) instrument (see Figure 2).

The polarization resistance *R_p_* is obtained from the electrochemical workstation via a three-electrode system (Figure 1b) by using linear polarization measurement, in which the working electrode is the steel wire immersed in the solution, the auxiliary electrode (counter) is the platinum electrode, and the reference electrode is the saturated Calomel/KCl electrode (SCE). The parametric values used for the linear polarization measurement are as follows. The electro-potential range for scanning is taken as ±10 mV relative to that of the working electrode. The scanning speed is 0.1667 mV/s. A total of 201 points are scanned. The data obtained are recorded and processed using Versastudio software (Princeton Applied Research, Oak Ridge, TN, USA). The polarization resistance *R_p_* is then calculated by the ratio of the slope of E-I curve at zero-potential point to the exposed area of the wire.

The visual inspection is carried out by using 3R Anyty high-resolution portable microscope (Anyty, Tokyo, Japan), which has camera and video functions on the amplified object. The use of the portable microscope not only overcomes the inaccuracy of the traditional artificial visual inspection on the rust product of the wire, but also enables the evolution of the wire from corrosion initiation to final break to be recorded. In the present experiment the magnification rate of 30 times is used. As an example, Figure 3 shows the typical images of the steel wire at the four different stages: (a) immediately after it is immersed into the solution, (b) corrosion is just initiated, (c) corrosion is in development, and (d) wire breaks. The details about when and under what experimental conditions these photos were taken are given in Figure 4, respectively.

As the objective of the present experimental study is to investigate the corrosion initiation and corresponding corrosion behavior of the reinforcing steel in concrete, various simulated concrete pore solutions with different Cl^−^ and/or OH^−^ concentrations are employed as the corrosive solution used in the experiments. Table 2 gives the details of the solutions used, in which the three solutions represent the control solution (pH = 7), completely carbonated solution (pH = 8.3), and partially carbonated solution (pH = 10.3), respectively [31,32]. In addition, different Cl^−^ concentrations are also used in the solutions of the same pH value in order to examine the effect of Cl^−^ on the corrosion process of the steel wire.

## 4. Experimental Results

### 4.1. Polarization Resistance Measurement

In order to make a comparison between the present experimental method and other methods, polarization resistance is measured during the immersion tests of the bow shaped device in the simulated concrete pore solution of pH = 10.3 and Cl^−^ = 0.025 mol/L. Before the test the steel wire has undergone a passivation treatment for seven days. The pre-tensile force used for the test is *F* = 42 N. The polarization resistance is measured between the steel wire and a counter electrode (see Figure 1b). Figure 4 shows the variation of the *R_p_* with the time obtained from one of the conducted 15 tests. It can be seen from the figure that there is an abrupt change in *R_p_* response curve around the time *t* = 45 min. This time represents the breakdown time of the passivation film surrounding the steel wire. Before that time the steel wire is protected by the passivation film, which makes the corrosive reactions difficult to take place and thus results in a high polarization resistance. However, after the passivation film is destroyed, the polarization resistance decreases rapidly, and thus the corrosion starts.

A careful examination on the steel wire surface during the polarization resistance test shows that, after the breakdown of the passivation film the corrosion in the steel wire starts. The visible corrosion of the steel wire is observed at the time of about *t* = 80 min (Figure 3b). The corrosion continues until to the time *t* = 990 min at which the steel wire breaks (Figure 3d) due to the corrosion-induced reduction of the cross-section area of the steel wire

Figure 5 shows the polarization curves obtained experimentally at four different stages, which represent the corrosion level of the wire at the initial stage of immersion, the stage just after depassivation, the stage when the rust is visually detected, and the stage after final break. The polarization curves were obtained from the electrochemical workstation via a three-electrode system (Figure 1b) by using linear polarization measurement, in which the working electrode is the steel wire immersed in the solution, the counter electrode is the platinum electrode, and the reference electrode is the saturated Calomel/KCl electrode (SCE). The corrosion potential values of these four stages are found to be −0.162 V, −0.322 V, −0.361 V, and −0.418 V, respectively. It can be seen from the figure that the corrosion current has a jump from the initial stage of immersion to the depassivation stage, whereas after the depassivation, the increase of the corrosion current becomes rather slow, representing the gradual development of the corrosion.

Figure 6 shows the breakdown time of the passivation film obtained from the polarization resistance measurement, the corrosion starting time visibly observed during the test, and the final break time of the steel wire obtained from the 15 immersion tests under identical conditions. It can be seen from the figure that, overall, the passivation breakdown time obtained from the polarization resistance method is close to the corrosion starting time visibly observed in the test and they also have the similar trend; whereas the break time of the steel wire is found to vary between specimens. The correlation analyses between these three times are shown in Figure 7. It can be seen from the figure that there is a good correlation between the breakdown time of the passivation film and the corrosion starting time visibly observed with the correlation coefficient of 0.96. In contrast, the correlations between the breakdown time of the passivation film and the break time and between the corrosion starting time visibly observed and the break time are not very good, with the correlation coefficients of 0.66 and 0.67, respectively.

### 4.2. Effect of Prestress of Steel Wire

The effect of the prestress of the steel wire on its corrosion is investigated by using the immersion tests of the bow shaped device in the simulated concrete pore solution of pH = 7.0 and Cl^−^ = 0.0125 mol/L. Before the test the steel wire has undergone a passivation treatment for seven days. The pre-tensile forces used in the tests vary from *F* = 7.31 N to *F* = 42.3 N, which are equivalent to the prestress level from 0.162*σ*_ult_ to 0.938*σ*_ult_.

Figure 8 and Figure 9 show the results of the corrosion starting time (*t_s_*) visibly observed during the tests and the final break time (*t_r_*) of the steel wire, respectively, obtained from 18 immersion tests with different pre-tensile forces applied in the tested steel wire. It can be seen from Figure 8 that, except for one test with *F* = 26.95 N, all other tests have very similar corrosion starting times with a mean value of 9.24 min and a standard deviation of 1.03 min. This indicates that the prestress of the steel wire has almost no influence on the chemical degradation process of the passivation film. This seems understandable. The time required for corrosion imitation is a chemical process during which the passivation surrounding the steel is gradually destroyed. Unlike the corrosion starting time, however, the break time of the steel wire is found to be closely related to the applied pre-tensile force (see Figure 9). The larger the applied pre-tensile force, the shorter the break time of the steel wire. This seems to be expected. As the applied pre-tensile force increases, the stress will be close to the ultimate strength of the steel, thus the allowable reduction of the cross-section area becomes small. As demonstrated by Equation (3), when *F*→*F*_max_ the break time of the steel wire will be very close to the corrosion starting time *t_r_*→*t_s_*; meaning that any tiny corrosion in the steel wire could lead to a break of the wire.

For the convenience of presentation, Equation (3) is rewritten as follows,
∆*F* = (*σ*_ult_*v*)∆*t*(5)
where ∆*t* = *t_r_*_−_*t_s_* is the time interval between the break time and corrosion starting time of the steel wire and ∆*F* = *F*_max_−*F* is the allowable force for the steel wire to break. Figure 10 shows the plot of ∆*t* versus ∆*F* for the above 18 immersion test data. It is observed from the figure that the relationship between ∆*F* and ∆*t* is almost linear, indicating that for a given simulated solution the reduction rate of the cross-section area is almost constant. Note that although ∆*t* increases with ∆*F*, the actual values of ∆*t* for most cases are not very big. For example, for the present 18 immersion tests, the maximum value of ∆*t* is about 420 min, which is only about 7 h. This value, when compared to the design life of a structure that is tens of years, is negligible. This indicates that the break time of the steel wire obtained from the present immersion tests can practically be taken as the corrosion starting time.

### 4.3. Effect of Passivation Film

The effect of the passivation of steel wire on the corrosion-induced break time (*t_r_*) of the steel wire is investigated by using the immersion tests of the bow shaped device in the simulated concrete pore solution of pH = 7.0 and Cl^−^ = 0.0125 mol/L. A total of 33 (3 × 11) tests are carried out, in which three tests are for the steel wire that has no passivation and the other 3 × 10 tests are for the steel wire that has undergone a passivation treatment in a saturated Ca(OH)_2_ solution with pH value of 12.5 using different treatment periods. The pre-tensile force used in the tests is *F* = 30 N. Figure 11 shows the break time of the steel wire obtained from the 33 tests. It can be seen from the figure that the results obtained from any three repeating tests are close, indicating that the test is rather reliable. Interestingly, the longest break time is found in the test where the steel wire has no passivation treatment, followed by the test where the steel wire has only one day passivation treatment. The break times for the steel wire that has passivation treatment between two and 11 days are not very different. While from the tendency of the variation of the curve shown in the figure it seems that the further increase of the passivation treatment period can slightly increase the break time.

To further confirm the finding that the passivation of the steel wire does not increase its corrosion break time, the immersion tests of the bow shaped device in the simulated concrete pore solutions of varying Cl^−^ concentration are also carried out. The steel wire used in the tests includes one without passivation treatment and one with passivation treatment for seven days. The pH value of the simulated solution is kept to be 7.0 for all of the tests. The pre-tensile force used in the tests is *F* = 30 N. Figure 12 shows the comparison of the break times of the steel wires with and without passivation for eight different Cl^−^ concentration solutions. It is evident from the figure that the passivation has a negative influence on the break time of steel wire disregarding the Cl^−^ concentration in the solution. This indicates that, although the passivation film can generally protect the steel from corrosion and thus can postpone the corrosion starting time, it may accelerate the corrosion process after the corrosion starts.

In order to explain why the passivation film has a negative effect on the break time of the steel wire, SEM analysis on the steel wires with and without passivation is carried out. Figure 13 shows the SEM images of the steel wires before they are immersed into the simulated concrete pore solution. It can be seen from the figure that the surface of the steel wire without the passivation is very uniform and smooth; whereas the surface of the steel wire with passivation is not very smooth and it looks like wearing a “knitting cloth”. The thickness and mesh density of the “knitting cloth” increase with the passivation time.

Figure 14 shows the SEM images of the corresponding steel wires after three-hour immersion in the simulated concrete pore solution of pH = 7.0 and Cl^−^ = 0.0125 mol/L. It can be seen from the figure that the corrosion of the steel wire without passivation is almost uniform, whereas the corrosion in other three steel wires that have had passivation treatment exhibits the pitting pattern.

### 4.4. Effect of Cl^−^ and OH^−^ Concentrations

The effect of the simulated concrete pore solution on the corrosion starting time (*t_s_*) and corrosion-induced break time (*t_r_*) of the steel wire is investigated by using the immersion tes *t_s_* of the bow shaped device in the simulated concrete pore solutions of different pH values and different Cl^−^ concentrations. Before the test the steel wire has undergone a passivation treatment for seven days. The pre-tensile force used in the tes *t_s_* is F = 30 N.

Figure 15 shows the variation of corrosion starting time with Cl^−^ concentration in three different types of simulated concrete pore solutions of pH = 7, 8.3, and 10.3, respectively. It can be seen from the figure that the corrosion starting time decreases with either the increase of Cl^−^ concentration or the decrease of pH value in the solution. This kind of results is expected since the higher the Cl^−^ concentration in the solution or the lower the pH value in the solution, the quicker the depassivation process of the steel will take place.

Figure 16 shows the variation of corrosion-induced break time of steel wire with Cl^−^ concentration in the three different types of simulated concrete pore solutions of pH = 7, 8.3 and 10.3, respectively. It can be seen from the figure that the wire break time decreases with the increase of Cl^−^ concentration but turns to be stable or even slightly recovered after the Cl^−^ concentration increases to a certain level. The break time regaining at high Cl^−^ concentrations is probably due to the corrosion product which, at a certain degree, provides a coating function to protect the steel from the chloride attack, and thus reduce the corrosion speed of the steel wire. The decrease of pH value in the solution also leads to a decrease of the wire break time.

## 5. Conclusions

In this paper a new mechanical-based experimental method has been proposed to investigate the corrosion initiation time and corresponding corrosion behavior of reinforcing steel in simulated concrete pore solutions. The experiment proposed uses a bow shaped device with a prestressed steel wire, which is immersed in a simulated concrete pore solution for corrosion test. The corrosion initiation of the steel wire during the immersion can be observed visibly. From the analysis of the obtained test results, the following conclusions can be drawn:

1. The proposed bow shaped device can be used to determine the corrosion initiation of reinforcing steel under the action of various different simulated concrete pore solutions and the corrosion rate of steel under different corrosion conditions.

2. The corrosion starting time obtained from the present experimental method is comparable with that obtained using the polarization resistance method for depassivation measurement.

3. The passivation film on the steel surface can protect steel from corrosion and thus postpone the corrosion initiation. However, in terms of the corrosion rate, the passivation film has a negative effect. It causes the steel to have pitting corrosion and can accelerate the corrosion process after the corrosion starts.

4. The pre-tensile force in the steel wire can affect the break time of corroded steel but has almost no effect on the corrosion initiation. The closer the pre-tensile force to the ultimate tensile strength of the uncorroded steel wire, the obtained break time of the corroded steel wire will be closer to the corrosion initiation time.

5. Both the chloride concentration and pH value in a simulated solution can affect the process of depassivation of the steel wire. The higher the chloride concentration or the lower the pH value, the quicker the depassivation of the steel wire to take place.

In general, the reduction rate of the cross-section area of the steel wire increases with the chloride concentration or a decrease of pH value in the solution. However, for the case where the chloride concentration is very high and the pH value is small, there is a slight decrease in the corrosion rate due to the coating function of the corrosion products surrounding the wire.

## Figures and Tables

**Figure 1 materials-13-05327-f001:**
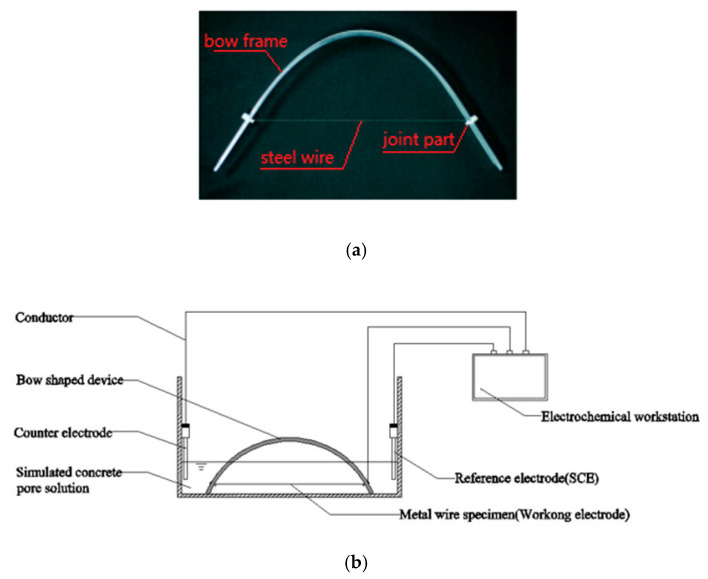
Illustration of corrosion test of steel wire in a corrosive solution. (**a**) Image and (**b**) schematic of corrosion test of steel wire in a corrosive solution.

**Figure 2 materials-13-05327-f002:**
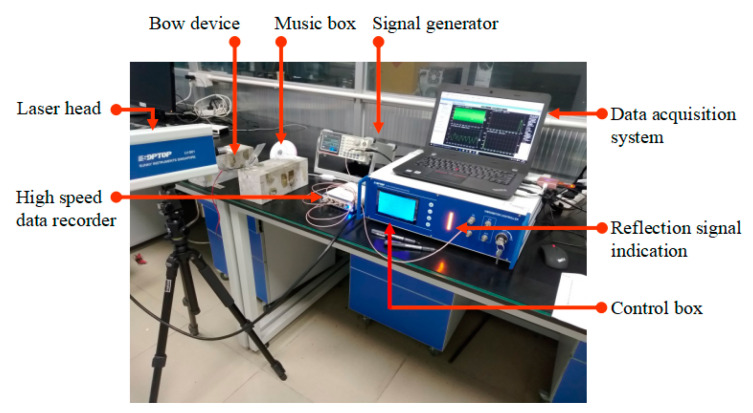
Frequency measurement of steel wire in bow shaped device using LDV.

**Figure 3 materials-13-05327-f003:**
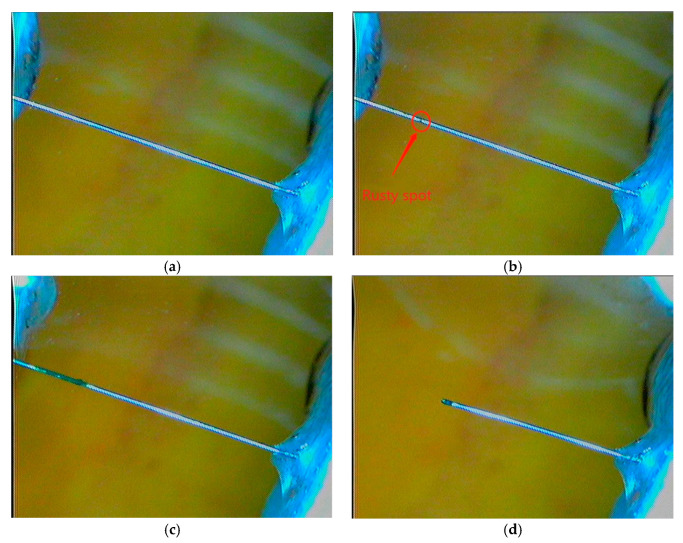
Images of steel wire at different stages. (**a**) Immediately after immersed into solution, (**b**) corrosion just initiated where a rust spot was viewed, (**c**) corrosion in development where corrosion region has been developed, and (**d**) after wire breaks (The details about when and under what experimental conditions these photos were taken are given in Figure 4, respectively).

**Figure 4 materials-13-05327-f004:**
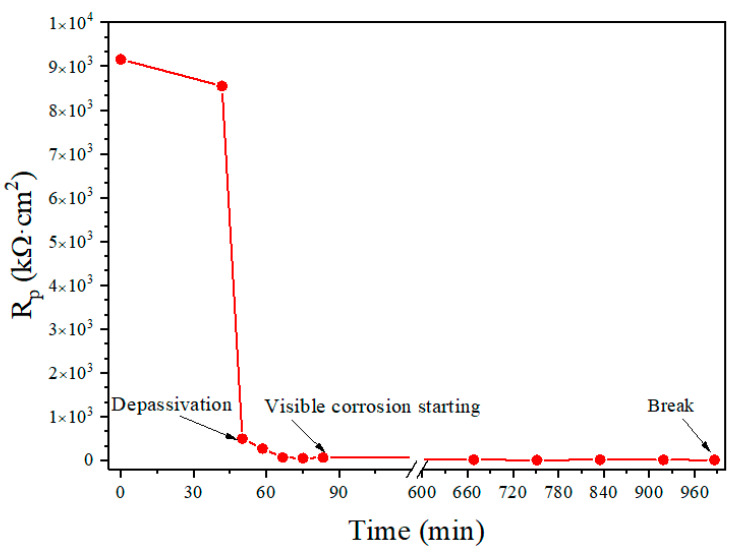
Time-history of *R_p_* value and key time points for depassivation, corrosion initiation and final break (*F* = 42 N, 7-day passivation, Cl^−^ = 0.025 mol/L, pH = 10.3).

**Figure 5 materials-13-05327-f005:**
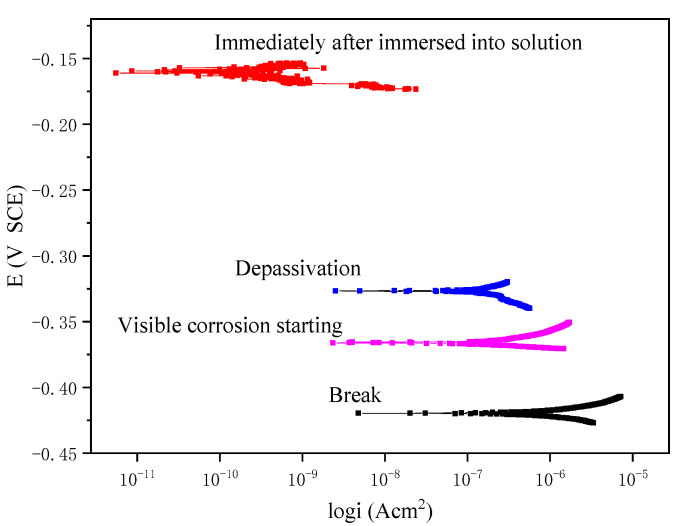
Polarization curves at different stages.

**Figure 6 materials-13-05327-f006:**
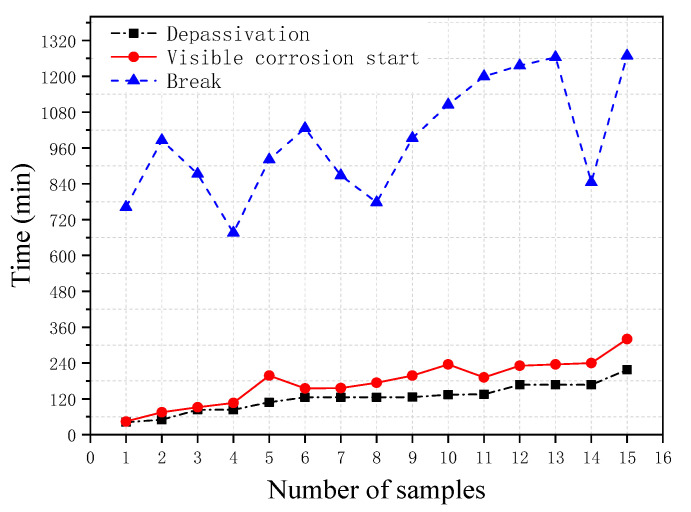
Depassivation time, corrosion starting time, and wire break time obtained from 15 corrosion tests (*F* = 42 N, 7-day passivation, Cl^−^ = 0.025 mol/L, pH = 10.3).

**Figure 7 materials-13-05327-f007:**
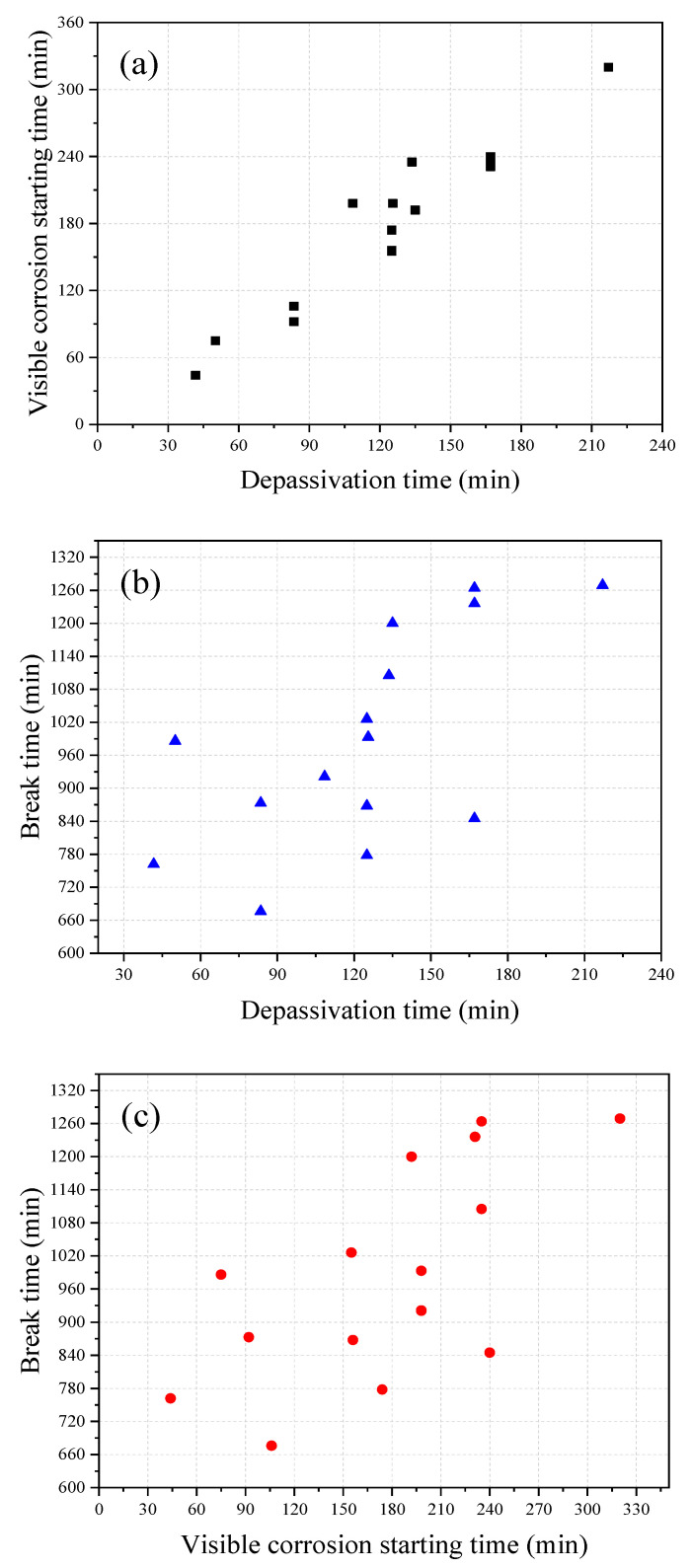
Correlation between (**a**) passivation breakdown time and visible corrosion starting time (correlation coefficient of 0.96), (**b**) between steel wire break time and passivation breakdown time (correlation coefficient of 0.66), (**c**) between steel wire break time and visible corrosion starting time (correlation coefficient of 0.67) obtained from 15 corrosion tests.

**Figure 8 materials-13-05327-f008:**
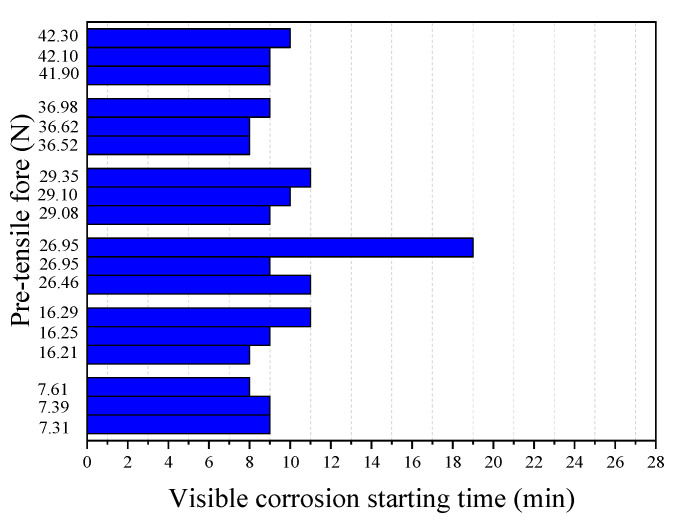
Corrosion starting times at different pre-tensile forces (pH = 7, Cl^−^ = 0.0125 mol/L).

**Figure 9 materials-13-05327-f009:**
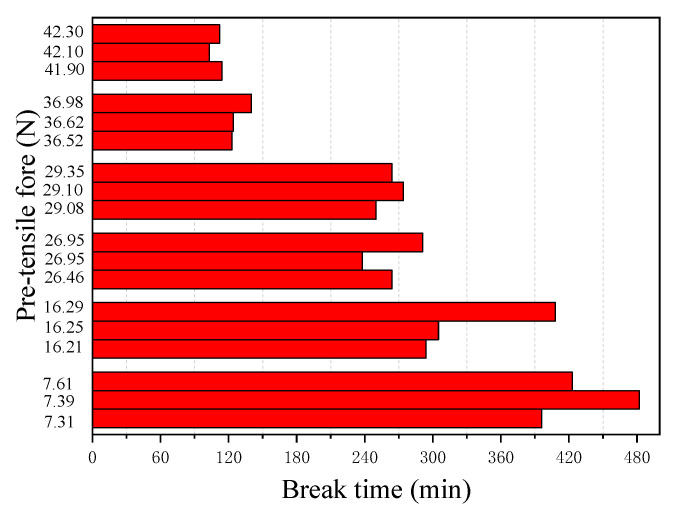
Steel wire break times at different pre-tensile forces (pH = 7, Cl^−^ = 0.0125 mol/L).

**Figure 10 materials-13-05327-f010:**
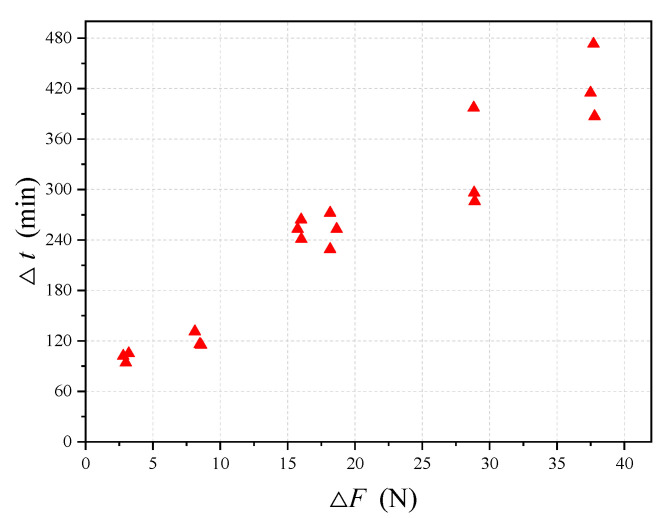
Relationship between ∆*t* and ∆*F* obtained from 18 corrosion tests.

**Figure 11 materials-13-05327-f011:**
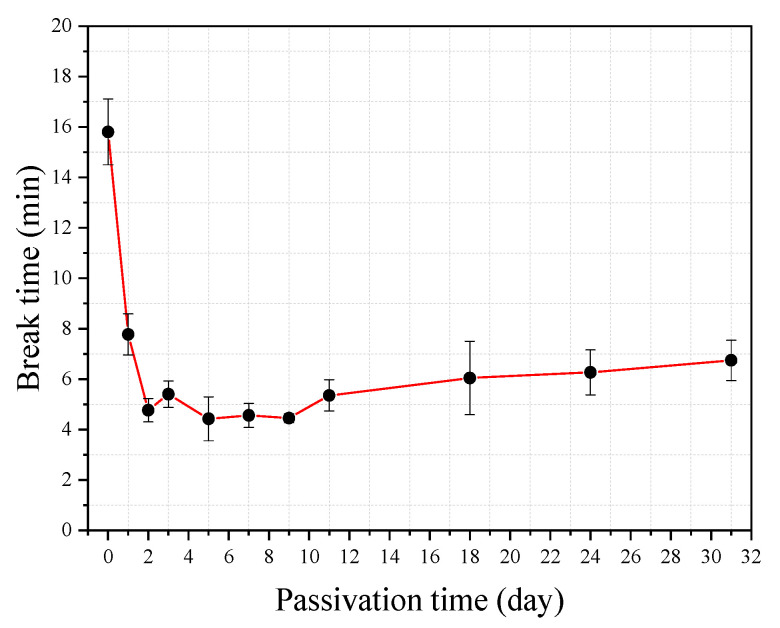
The break time of steel wire with and without passivation treatment (pH = 7, *F* = 30 N, Cl^−^ = 0.0125 mol/L).

**Figure 12 materials-13-05327-f012:**
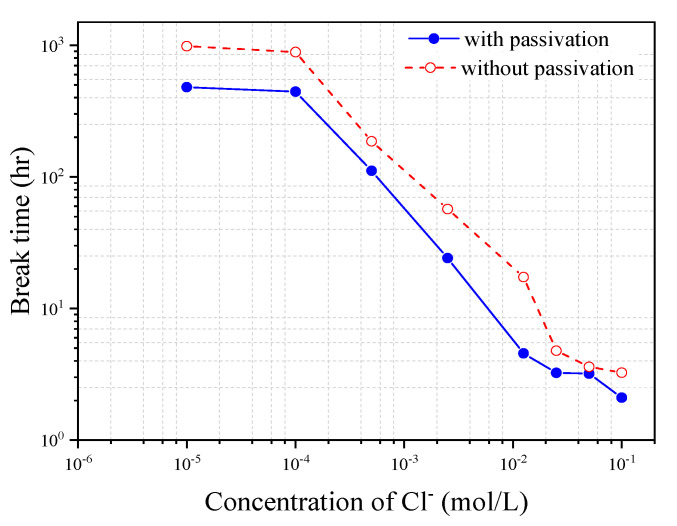
The break time of steel wire with and without passivation treatment under different Cl^−^ concentration solutions (pH = 7, *F* = 30 N).

**Figure 13 materials-13-05327-f013:**
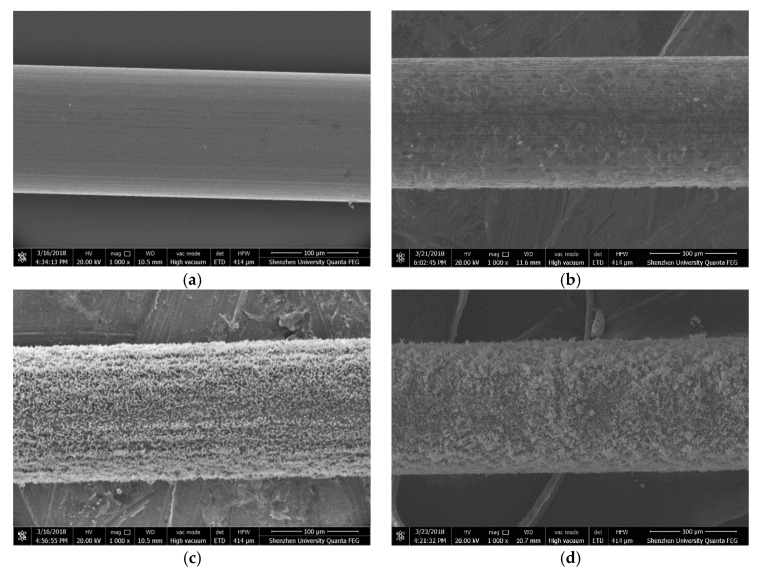
SEM images of steel wires, (**a**) without passivation, (**b**) 1-day passivation, (**c**) 7-day passivation, and (**d**) 11-day passivation.

**Figure 14 materials-13-05327-f014:**
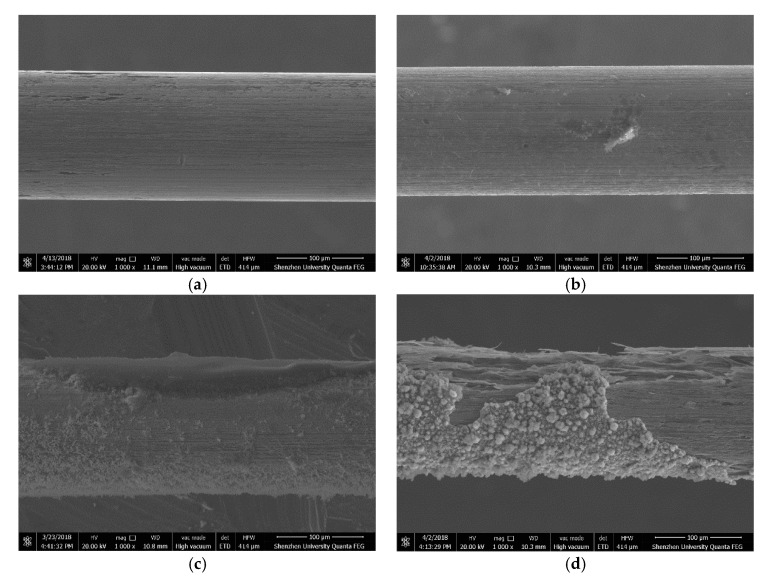
SEM images of corroded steel wires, (**a**) without passivation, (**b**) 1-day passivation, (**c**) 7-day passivation, and (**d**) 11-day passivation.

**Figure 15 materials-13-05327-f015:**
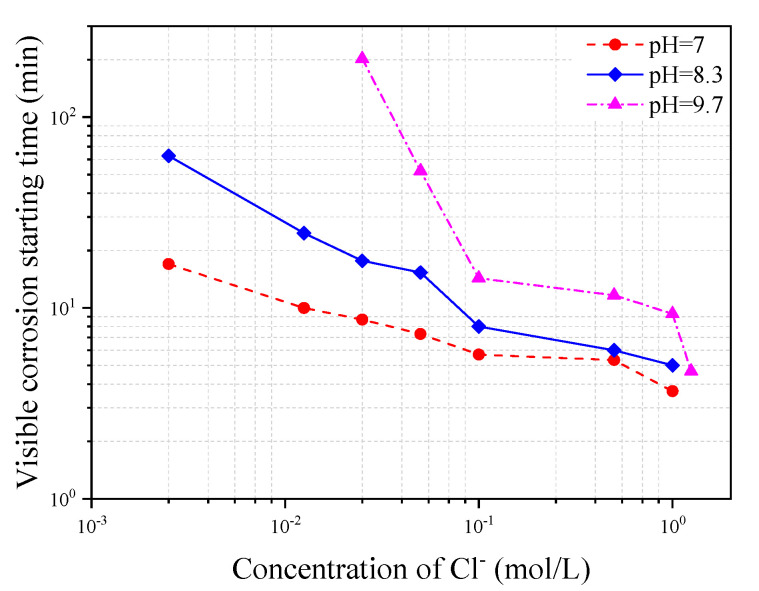
Variation of corrosion starting time with chloride concentration in three different types of simulated concrete pore solutions.

**Figure 16 materials-13-05327-f016:**
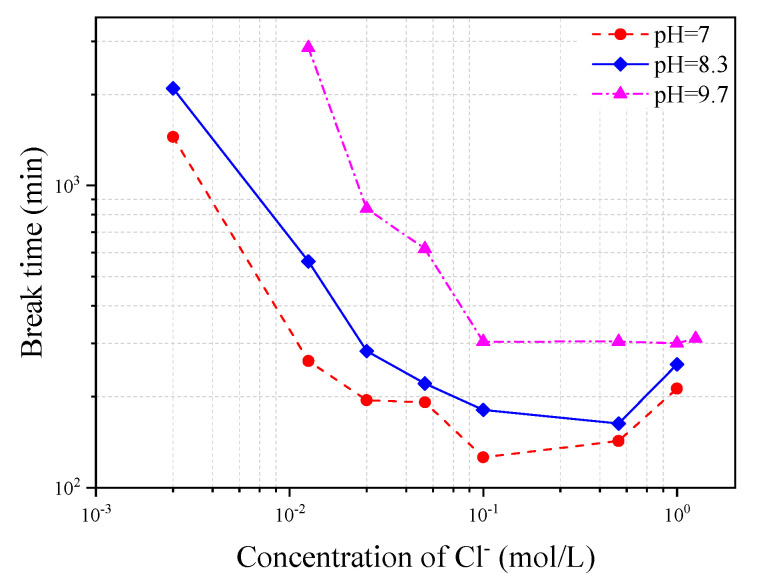
Variation of steel wire break time with chloride concentration in three different types of simulated concrete pore solutions.

**Table 1 materials-13-05327-t001:** Chemical compositions of steel wire (% in mass).

Sample Points	C	O	Na	P	Mn	Co	Zn	Si	Fe
1	3.91	7.85	0.04	2.84	1.03	1.47	5.14	0.29	77.43
2	4.31	4.32	0.05	1.5	1.18	1.39	2.82	0.87	83.56
3	4.62	8.48	0.06	3.07	1.67	1.28	5.89	0.55	71.55
4	6.16	1	0.05	0.53	2.61	2.82	3.53	0.86	82.43
5	6.72	5.82	0.05	2.48	2.17	2.19	7.42	0.34	72.81

**Table 2 materials-13-05327-t002:** Details of simulated concrete pore solutions used in experiments.

Type of Solution	pH	OH^−^ mol/L	CL^−^ mol/L
Deionized H_2_O	7	1 × 10^−7^	0, 0.00001, 0.0001, 0.005, 0.0025, 0.0125, 0.025, 0.05, 0.1, 0.5, 1.0
Saturated CaCO_3_	8.3	1 × 10^−5.7^	0.0025, 0.0125, 0.025, 0.05, 0.1, 0.5, 1.0
0.53 g/L NaHCO_3_ + 1.26 g/L Na_2_CO_3_	10.3	1 × 10^−4.3^	0.0125, 0.025, 0.05, 0.1, 0.5, 1.0, 1.25

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
