# Peer review of "A Novel Testing Method for Examining Corrosion Behavior of Reinforcing Steel in Simulated Concrete Pore Solutions"

_materials, 2020, doi:10.3390/ma13235327_

Round 1
Reviewer 1 Report
In this paper, the authors describe a bow-like steel wire that can be employed to determine the corrosion behaviour of steel in simulated concrete solutions. This is a simple but interesting approach, however I recommend some changes before this paper is considered for publication. The authors should provide more details on the previous literature in this area and clearly show how their method is different. Also, the authors should refer to some of the many review articles that have been devoted to advances in the corrosion monitoring of reinforced concrete structures.
On Page 2, Line 55, the authors state that ‘most of these methods do not have good repeatability (electrochemical methods). However, the authors do not mention repeatability elsewhere. There is no indication of reproducibility in any of the figures, for example, Figures 4, 6, 7, 10 and 12 would benefit from having error bars or some other details on reproducibility. Moreover, there appears to be considerable variation in the break times in Figure 6 and some comment on how this compares to the repeatability of the electrochemical based measurements should be given.
On page 4,Line 118, the authors state that the ends of the wires are coated with epoxy resin. How do the authors avoid the formation of crevice corrosion at this epoxy interface? Or are they confident that crevice corrosion does not occur meaning that all the electrochemical data are free from crevice corrosion events.
On Figure 5, polarisation data are shown for the ‘break’ electrode. The authors should provide more details on how this measurement is made. Why does the break in the wire not break the circuit and give no current flow?
The data in Figures 8 and 9 require further explanation. If the final aim is to use the breakage of the wire to estimate the onset of corrosion, then why does the break time depend on the force, but the onset of corrosion is largely independent of the force?
The authors should provide some details on how this approach could be employed to monitor the corrosion of steel in concrete.
Finally, there are some typographical errors that need to be corrected.
Reviewer 2 Report
1. Summary
It should describe what was done in the experiment and what main results were obtained.
2. Introduction
Each publication should be discussed in more detail. It is not enough to list a few in one reference. Finally, briefly describe what is presented in this publication.
3. Method and device, and 4. Experimental Program
- The physico-chemical properties of the wire should be described in more detail. What was the material grade - ISO designation? What was the dimension? How the tension in the wire was controlled. How was it starting? Has it changed during the research? Table 1 shows the mass shares in points - what?
- Was the change in stress over time controlled - if there was any property of the section due to corrosion?
- How was the corrosive environment established. Reaction and electronegativity?
- Figure 3 is illegible. Nothing significant for publication as it stands. For what times or specific test conditions was the wire photographed? This information should be included in the figure caption.
- The places, times for which characteristic measurements and photos were taken, should be marked on diagram 4.
- Please mark the correlation coefficient on the graphs and calculate the mean value and standard deviation.
- Please describe the scatter in the results in detail in figure 9.
- SEM photos are not clear. What scale and measurement parameters are they? Please describe and mark the characteristic areas on the wire surface.
4. Conclusion
The conclusions are too general. As a result of the research, specific results were obtained. Qualitative and quantitative assessment has been made - please refer to the results and certain mechanisms of the corrosion process.
Round 2
Reviewer 1 Report
The authors have made the recommended changes and the paper is now clearer, providing a more relevant introduction section, more details on reproducibility and I recommend that the paper be accepted.
Reviewer 2 Report
Thanks for comments. The main editor makes the final decision.